# Prognostic significance of CD8+ T cell Spatial Biomarkers in ER+ and ER− breast cancer: A retrospective cohort study

Andrew E. Walker[1]*, Xiaohua Gao[2], Qichen Wang[3], Gabriela De la Cruz[2], Didong Li[4], Charles M. Perou[2,5,6], Joel Saltz[7], James Steve Marron[1,2,4‡], Katherine A. Hoadley[2,5‡], Melissa A. Troester[2,3‡]

1 Department of Statistics and Operations Research, University of North Carolina, Chapel Hill, North Carolina, United States of America, 2 UNC Lineberger Comprehensive Cancer Center, University of North Carolina, Chapel Hill, North Carolina, United States of America, 3 Department of Epidemiology, University of North Carolina, Chapel Hill, North Carolina, United States of America, 4 Department of Biostatistics, University of North Carolina, Chapel Hill, North Carolina, United States of America, 5 Department of Genetics, University of North Carolina, Chapel Hill, North Carolina, United States of America, 6 Department of Pathology and Laboratory Medicine, University of North Carolina, Chapel Hill, North Carolina, United States of America, 7 Department of Biomedical Informatics, Stony Brook University, Stony Brook, New York, United States of America

‡ These authors are *co-senior authors on this work.*
* awalker@unc.edu

## Abstract

### Background

Tumor infiltrating lymphocytes (TILs) are prognostic in triple-negative breast cancer, but not estrogen receptor (ER) positive cancers which comprise 70%–80% of breast cancers. This is due to the relatively low immune infiltration in ER-positive tumors. However, few studies have explored the prognostic impact of lower abundance TILs evaluated using spatial methods. The objective of this study was to explore whether the distribution of lymphocytes with respect to tumor cells predicts prognosis.

### Methods and findings

In this retrospective cohort study, we used multiplex immunofluorescent (IF)-stained images of tissue microarray cores (stained for cytokeratin [Ck], CD8, and FoxP3) obtained from 1,467 study participants to compute distance-based visual morphometry for epithelial and immune cells, including two new metrics, *proximity* and *consistency*. Proximity and consistency are defined as functions of the mean and variance of nearest neighbor distances between Ck+ tumor cells and CD8+ T cells. Prognostic significance of proximity and consistency were compared to lymphocyte counts using log-rank tests of differences in Kaplan–Meier survival curves and Cox proportional hazards models. Better recurrence-free survival (RFS) was observed for both ER+ and ER− breast cancers with high proximity and consistency of CD8+T cells. Among ER− breast cancers, proximity

**Data availability statement:** The participant data used in this study are considered "sensitive" according to NIH policies. As such, part of the UNC IRB approval for this study requires a data use agreement for data sharing. The Carolina Breast Cancer Study has a Letter of Intent process through the study website (https://unclineberger.org/cbcs) for data sharing requests.

**Funding:** This research was supported by a grant from UNC Lineberger Comprehensive Cancer Center, which is funded by the University Cancer Research Fund of North Carolina, the Susan G Komen Foundation (OGUNC1202, OG22873776, SAC210102, TREND21686258 to MAT and AEW), National Cancer Institute (R01CA253450 to MAT and KAH), the National Cancer Institute Specialized Program of Research Excellence (SPORE) in Breast Cancer (NIH/NCI P50-CA058223 to CMP, MAT, KAH, and JSM), the Breast Cancer Research Foundation (HEI-23-003 to CMP and MAT), and the US Department of Defense (HT94252310235 to MAT). This research recruited participants and/or obtained data with the assistance of Rapid Case Ascertainment (RCA), a collaboration between the North Carolina Central Cancer Registry and UNC Lineberger. RCA is supported by a grant from the National Cancer Institute of the National Institutes of Health (P30CA016086 to CMP, MAT, KAH, and JSM). The funders had no role in study design, data collection and analysis, decision to publish, or preparation of the manuscript.

**Competing interests:** I have read the journal's policy, and the authors of this manuscript have the following competing interests: C.M.P is an equity stockholder and consultant of BioClassifier LLC; C.M.P is also listed as an inventor on patent applications for the Breast PAM50 Subtyping assay.

**Abbreviations:** AE, adaptive-enriched; AGI, any genomic instability; CBCS, Carolina Breast Cancer Study; CK, cytokeratin; ER, estrogen receptor; FFPE, formalin-fixed, paraffin-embedded; HE, hematoxylin–eosin; HER2, human epidermal growth factor receptor 2; HR, hazard ratio; HRD, high homologous recombination deficiency; IE, innate-enriched; IF, immunofluorescent; NGI, no genomic instability; OR, Odds Ratio; PR, progesterone receptor; REMARK, REporting recommendations for tumor MARKer; RFS, recurrence-free survival; TIL, tumor infiltrating lymphocyte; TNBC, triple-negative breast cancer; UNC, University of North Carolina.

had the highest RFS hazard ratio (HR 1.84, 95% CI [1.18, 2.87]; $p=0.0069$) compared to count and consistency. Among ER-positive participants, RFS hazard ratios for proximity and consistency were 2.04 (95% CI [1.39, 2.98]; $p=0.0003$) and 1.82 (95% CI [1.23, 2.69]; $p=0.0026$), respectively. These associations were stronger than those observed for lymphocyte count (HR 1.35, 95% CI [0.92, 1.98]; $p=0.1289$). Independent prognostic value was demonstrated by controlling clinical and demographic variables such as age, tumor grade, stage, ER status, progesterone receptor status, and human epidermal growth factor receptor 2 status. These IF-derived spatial metrics were also associated with established TILs metrics and RNA expression-based measures of tumor adaptive immune response. Though these results are promising, our exploration of the tumor immune microenvironment was limited by the small number of immune markers available for our data.

## Conclusions

Spatial characteristics described by proximity and consistency are frequently associated with recurrence irrespective of ER status. The prognostic significance of proximity in ER+ breast cancer implies that spatial parameters may identify individuals who would benefit from immune therapy; up to 75% of breast cancers experience T cell proximity suggestive of immune susceptibility.

## Author summary

### Why was this study done?

- Immune cells present in tumors are predictive of survival outcomes in certain subtypes of breast cancer.

- Previous studies have not established clear associations between immune cell presence and survival in estrogen receptor positive breast cancers, a common subtype where immune cells are less abundant.

- Spatial methods that are independent of the number of immune cells have not been thoroughly studied but could provide insights into breast cancers where immune cells are less abundant.

### What did the researchers do and find?

- We identified immune cells and tumor cells present in images of breast cancer tissue.

- We developed two spatial measurements based on the distance between CD8+T cells and tumor cells.

- Using these measurements, we showed that proximity of CD8+T cells to tumor cells was associated with better survival outcomes, even in estrogen receptor positive breast cancer.

> ### What do these findings mean?
>
> - Even in low abundance, the presence of immune cells could indicate a better chance of survival.
>
> - Immunotherapy may benefit a broader array of patients with breast cancer who were previously ineligible.
>
> - A key limitation of this study is that we were unable to fully interrogate immune cell interactions within the tumor due to a lack of available markers.

## Introduction

The presence of tumor-infiltrating lymphocytes (TILs) is prognostically significant in breast cancer [1–3]. In addition, immune therapy in conjunction with chemotherapy has been successful in treating triple-negative breast cancer (TNBC) [4,5]. However, 70%–80% of breast cancers are estrogen receptor (ER+) and not eligible for immunotherapy under current guidelines [6–8]. Many of these tumors often have moderate immune infiltration. Recently, several trials demonstrated potential benefit of immune checkpoint inhibitors in high-grade ER+/human epidermal growth factor receptor (HER2−) breast cancer [9,10]. Given the ability to target the tumor immune microenvironment and the potential benefits of immune therapy in combination with adjuvant chemotherapy, there is a need to better understand immune cell interactions with tumors across breast cancer subtypes [5,11].

Our previous work linked genomic instability with immune response in both ER+ and ER− participants, suggesting that immunogenic genomic instability is prognostically significant even in ER+ participants [12]. However, immunogenic responses are difficult to detect in ER+ breast cancers as the TIL counts tend to be present at low or moderate levels. Therefore, it may be important to consider whether immune cells are in close proximity to the tumor cells. In context of the high-count immune infiltration of ER− breast cancers, the assumption of proximity is more readily met. However, in moderately infiltrated tumors, distance between tumor and immune cells may vary. Very few studies have evaluated proximity [13,14] or other spatial characteristics [15,16], and few have used specific cell-type markers in analyses [11,17], such as those available via multiplex immunofluorescent (IF) [18]. Reliance on TILs counts from hematoxylin–eosin (HE) stained images or bulk gene expression [2,3,19] ignores the potentially important spatial relationship between CD8+T cells and tumor cell death [20].

We hypothesized that the distribution of lymphocytes with respect to tumor cells predicts prognosis. We considered proximity (measure of mean nearest neighbor distance between CD8+T cells and cytokeratin [Ck+] tumor cells) and consistency (measure of variability in nearest neighbor distance between CD8+T cells and Ck+ tumor cells) as two spatial metrics describing the distribution of CD8+T cells within the tumor immune microenvironment. We compared these measures to established gene expression and count-based measures for breast cancer immune microenvironments and evaluated associations with breast cancer prognosis in a diverse study population of participants with breast cancer.

## Methods

This study is reported as per the REporting recommendations for tumor MARKer prognostic studies (REMARK) guideline (S1 REMARK Checklist) [21].

### Study population

This research used pathologic images from breast cancers in Phase 3 (2008–2013) of the Carolina Breast Cancer Study (CBCS). Methods of data collection for Phase 3 CBCS have been previously described in detail [22]. Participants were recruited using rapid case ascertainment from the North Carolina Central Cancer Registry [23]. CBCS eligibility required participants to be females aged 20–74 years with their first diagnosis of invasive breast cancer, and they must be residents of specified counties in North Carolina [23]. Participants were enrolled under the University of North Carolina at

Chapel Hill School of Medicine Institutional Review Board protocol and provided informed written consent [23]. Multiplex IF images of tissue microarray cores, as well as time to recurrence data, were available for 1,740 participants. The dataset was reduced to 1,467 participants after removing tissue microarray cores which did not pass quality control standards, as described in the image processing section. A detailed quality control flow chart is provided in the supplemental material (S1 Fig). Distributions of common demographic and clinical variables were computed before and after quality control to ensure that reduction in participants did not introduce bias (Fig 1). We did not observe any substantial changes in the distribution of these variables. Of the 1,467 participants, 914 had Immune Class data [24], and 864 had Any Genomic Instability/No Genomic Instability (AGI/NGI) classes [12], and only 288 participants had whole slide image TIL spatial scores, including intratumoral strength, peritumoral strength, TIL forest, and TIL desert scores [25]. These RNA-based and larger-scale spatial metrics were compared to spatial metrics derived herein to facilitate biological interpretation.

## Tissue microarray staining and processing

Up to four 1 mm cores were cut from a formalin-fixed, paraffin-embedded (FFPE) tissue block and added to a tissue microarray grid, and tissues were obtained from a subset of CBCS3 cases. Details of CBCS3 tissue microarray construction have been previously described [26]. Triple immunofluorescence was performed on 5 μm FFPE sections using the Bond Rx fully automated slide staining system (Leica Biosystems) and Bond Research Detection kit (DS9455). Slides were deparaffinized in Leica Bond Dewax solution (AR9222), hydrated in Bond Wash solution (AR9590), and sequentially stained for CD8 (Cell Marque, 108R-14), FoxP3 (abcam, ab20034), and Ck (Leica Biosystems, NCL-L-AE1/AE3-601). Specifically, antigen retrieval for CD8 was performed for 20 min at 100 °C in Bond-Epitope Retrieval solution 1 (ER1), pH 6.0 (AR9961). After pretreatment, slides were incubated for 1 h with CD8 (1:200) at 25 °C, followed with Novolink Polymer (Leica Biosystems, RE7260-CE), then TSA Cy3 (Akoya Biosciences, SAT704A001EA). A second round of antigen retrieval was performed for 10 min at 100 °C in Bond-epitope retrieval solution 1 (ER1). Slides were then incubated with the FoxP3 (1:120, 2 h) at 25 °C, then Novolink Polymer or Post Primary plus Polymer combination and detected with TSA Cy5 (Akoya Biosciences, SAT705A001EA). A final round of antigen retrieval was performed with ER1 for 10 min before incubating the slides in Ck (1:300, 30 min) at 25 °C, followed by Novolink Polymer or Post Primary plus Polymer combination and Alexa Fluor 488 Tyramide Reagent (Thermo Fisher Scientific, B40953). Nuclei were stained with Hoechst 33258 (Invitrogen). The stained slides were mounted with ProLong Gold antifade reagent (Thermo Fisher Scientific, P36930). Positive and negative controls (no primary antibody) were included in this run. High-resolution acquisition of IF slides was performed with the Aperio Versa 200 scanner (Leica Biosystems) at an apparent magnification of 20×. Images were uploaded to the eSlideManager database (Aperio). The staining process was done by Pathology Services Core at University of North Carolina at Chapel Hill (UNC), and this study was approved by the UNC Office of Human Ethics and Institutional Review Board.

## Image processing

Multiplex IF images consisted of slides of multiple 1 mm tissue microarray cores, where most participants had up to four cores, and a small group of participants, who had two tissue samples taken, had up to eight cores. Tissue microarray cores were taken from a pathologist-identified intratumoral region selected for a high percentage of epithelium. The images had four color channels with each color indicating one of four biomarkers: Hoechst (blue), Cytokeratin (cyan), CD8 (green), and FoxP3 (red). The Hoechst channel is an indicator for the cell nucleus, CD8 and FoxP3 are indicators for lymphocytes, and Cytokeratin is an indicator for tumor cells.

Cell segmentation and classification were performed in Qupath digital pathology software [27]. Cells were segmented using the cell detection function in Qupath which segments the cells based on the intensity of the Hoechst channel. Pathologists visually assessed several samples to confirm that the final segmentation gave a reasonable representation. Cell classification was performed using the simple threshold classifier in Qupath. The cells were classified as tumor,

| | Initial Cohort | Final Cohort |
|---|---|---|
| | **N = 1,740**[1] | **N = 1,467**[1] |
| Age | | |
| <50 year | 800 (46%) | 712 (49%) |
| ≥ 50 years | 940 (54%) | 755 (51%) |
| RACE | | |
| African American | 840 (48%) | 733 (50%) |
| Non-AA | 900 (52%) | 734 (50%) |
| Grade | | |
| I | 368 (21%) | 266 (18%) |
| II | 651 (37%) | 525 (36%) |
| III | 716 (41%) | 672 (46%) |
| Unknown | 5 (0.3%) | 4 (0.3%) |
| Stage | | |
| I | 783 (45%) | 626 (43%) |
| II | 716 (41%) | 631 (43%) |
| III | 201 (12%) | 173 (12%) |
| IV | 40 (2.3%) | 37 (2.5%) |
| ER | | |
| Missing | 49 (2.8%) | 24 (1.6%) |
| Negative | 521 (30%) | 474 (32%) |
| Positive | 1,170 (67%) | 969 (66%) |
| PR | | |
| Missing | 48 (2.8%) | 24 (1.6%) |
| Negative | 683 (39%) | 474 (32%) |
| Positive | 1,009 (58%) | 969 (66%) |
| HER2 | | |
| Positive | 255 (15%) | 227 (15%) |
| Negative | 1,485 (85%) | 1,240 (85%) |
| [1] n (%) | | |

**Fig 1. Distribution of demographic and clinical variables for participants before and after quality control.** The table provides the distributions of Phase 3 Carolina Breast Cancer (CBCS) study participants with available multiplex immunofluorescent tissue microarray images for several important clinical and demographic variables, both before and after quality control. There are minimal changes to the distribution of the participants after quality control was applied. Number of participants (*N*); Non-African American (Non-AA); estrogen receptor (ER); progesterone receptor (PR); human epidermal growth factor receptor 2 (HER2).

CD8+T cells, CD8+/FoxP3+, or unclassified. Thresholds were chosen based on careful analysis of the pixel level average channel intensity within the cell boundary. The cells were classified as CD8+/FoxP3+ when the average intensity for both the CD8 and FoxP3 channel exceeded the threshold, CD8+T cells when the average intensity of the CD8, but not the FoxP3, exceeded the threshold, and tumor only when the average intensity for the Ck channel was above the threshold and below the threshold for FoxP3 and CD8. Pathologists visually assessed several sample images to validate the final classification. QuPath quantitation of CD8 was highly correlated with Halo quantitation on the same slides ($\rho = 0.97$). QuPath CD8 density measures (counts per unit area) were also correlated with RNA expression data from separate cores from the same tumor specimen ($\rho = 0.49$). Details on parameters used in the cell detection function, as well as the cell classification thresholds that were used, are available in the supplemental material (S1 Table). The centroid coordinates of the cells, as well as the class labels, were exported from QuPath, and the proximity and consistency biomarkers were computed in Python as discussed in the next section. To ensure the quality of the data, any tissue microarray cores with fewer than 1,000 tumor cells and fewer than 3,000 total cells were removed.

## Spatial biomarkers

The spatial biomarker values were computed in Python, and all spatial measurements were conducted prior to receiving clinical data. The nearest neighbor distance from a tumor cell to the CD8+T cell was computed for each tissue microarray core. All distances were less than 1,000 microns, based on the size of the tissue microarray cores. Due to the highly skewed distribution of the nearest neighbor distances, the base 10 logarithm (log10) of nearest neighbor distance was determined to be a better representation. The collection of log10 nearest neighbor distances for each tissue microarray core corresponding to a single study ID were aggregated, and the mean and variance for the distribution of aggregated log10 nearest neighbor distances was computed for each study ID. The consistency metric was interpreted as a measure of intraindividual and cross-core heterogeneity. The mean and variance of log10 nearest neighbor distances are not intuitive measurements of proximity to tumor cells and consistency of distances. Instead, proximity is computed as a function of the mean of log10 nearest neighbor distances, and consistency is computed as a function of the variance of log10 nearest neighbor distances. This ensures that high values of proximity imply that CD8+T cells are closer on average to tumor cells, and high values of consistency have less variability in the distances between CD8+T cells and tumor cells. However, interpretation of these metrics is relative and not an absolute distance metric.

Proximity was given by the formula: (3—mean of log10 nearest neighbor distances from tumor cell to CD8+T cell). Proximity can range from 0 to 3 since the maximum mean of log10 nearest neighbor distances is less than 3. Consistency was given by the formula: (1—variance of log10 nearest neighbor distances from tumor cell to CD8+T cell). Consistency ranges from 0 to 1 since the maximum variance of nearest neighbor distances is less than 1. The lymphocyte count was computed as the total number of cells which were classified as CD8+T cells and CD8+/FoxP3+cells. Lymphocyte count at the participant level is computed as the average of lymphocyte counts over all of the participants' cores. When participants were split based on high/low lymphocyte count, the median count was used as the cut point. Proximity and consistency were binarized for the purpose of comparing differences in recurrence between participants with high versus low values of proximity/consistency. We explored various cut points for proximity and consistency. The cut point used in the analysis was based on splitting the data into four groups based on the quartiles and then collapsing groups with similar survival. After collapsing, the median was the final cut point for consistency, and the 1st quartile was the final cut point for proximity. An attempt was made to find an optimal, data-driven cut point using a training and testing split, but this led to unstable results due to only a small fraction of participants having a recurrence. The results from the attempted optimization are provided as supplementary material (S2 Table).

## Statistical analysis

The Kruskal–Wallis test, a nonparametric version of the analysis of variance test, was used to determine whether there are statistically significant differences between the medians of the proximity, consistency, and log10 lymphocyte count

distributions by ER status, as well as among breast cancer subtypes. The Kaplan–Meier method was used to estimate the recurrence-free survival (RFS) curves. All curves include estimates of 95% confidence bounds. The log-rank test was used to compute the *p*-values for difference between Kaplan–Meier estimates of survival curves. The hazard ratios and corresponding 95% confidence intervals were computed using the Cox proportional hazards model. Continuous count and proximity measures were normalized before inclusion in Cox proportional hazards models. All *p*-values for multivariate survival models came from the Cox proportional hazards model. In all cases, the proportional hazards assumption was confirmed using the proportionality test described by Grambsch and Therneau (1994) [28] as implemented by the cox. zph function in the R survival package. The likelihood ratio test was used for determining significance when comparing models in the case where the set of covariates of one model is a subset of the more complex model. Spearman's correlation coefficient was used to determine monotonic association between our count, proximity, and consistency metrics and the intratumoral strength, peritumoral strength, TIL forest, and TIL desert scores. Fisher's z-transformation was used to estimate 95% confidence intervals for the Spearman's correlation coefficient. Odds Ratios (OR) and 95% confidence intervals were computed assuming the data was generated by a multinomial model to determine the level of association of proximity, consistency, and lymphocyte count with AGI classified participants and to determine the level of association of proximity, consistency, and lymphocyte count with Adaptive-Enriched (AE) participants. The 0.05 level is used for all tests of statistical significance. All statistical analysis was performed using SciPy version 1.14.1 in Python, except survival analysis. Kaplan–Meier curves were computed using Lifelines version 0.29.0 in Python, and the Cox proportional hazards models were computed using Survival 3.5.5 in R.

## Statement of ethics

The study was approved by the University of North Carolina Institutional Review Board (92-0410) in accordance with U.S. Common Rule. All study participants provided written informed consent prior to study entry. This study complied with relevant ethical regulations, including the Declaration of Helsinki.

## Results

### Proximity and consistency

To quantify lymphocyte infiltration, we computed proximity, a function of the mean nearest neighbor distance from each tumor cell to CD8+ T cells (Fig 2). CD8+ T cells were defined as CD8+/FoxP3−. Lymphocyte count, defined as number of CD8 or FoxP3 positive cells, was computed for comparison to proximity and consistency since count is a commonly used spatial metric.

We computed kernel density estimates of the distributions of proximity, consistency, and count (Fig 3). The distributions of proximity and count tended to be wider, while consistency had a much tighter distribution. The H-statistic given by the Kruskal–Wallis test was used to determine if there was evidence of differences between the median of the count, proximity, and consistency by ER status (Fig 3). High proximity, high consistency, and high count showed strong evidence of association with ER status. In terms of magnitude, the log10 lymphocyte count had the strongest association with ER status ($H = 76.44$, $p < 0.0001$), followed by proximity ($H = 17.45$, $p < 0.0001$) and consistency ($H = 6.01$, $p = 0.01$).

We also evaluated differences between the median of the count, proximity, and consistency by PAM50 molecular subtypes (Fig 3). Similarly, the difference was greatest for log10 lymphocyte count by PAM50 subtypes ($H = 92.85$; $p < 0.0001$), but differences were also significant for proximity ($H = 26.48$; $p < 0.0001$) and consistency ($H = 8.77$; $p = 0.03$). As with ER status, the distributions of proximity and consistency were overlapping, notably in Basal versus Luminal. This implies that proximity and consistency do not distinguish ER status and PAM50 subtypes as well as lymphocyte count, but rather they are spatial parameters that are shared across subtypes.

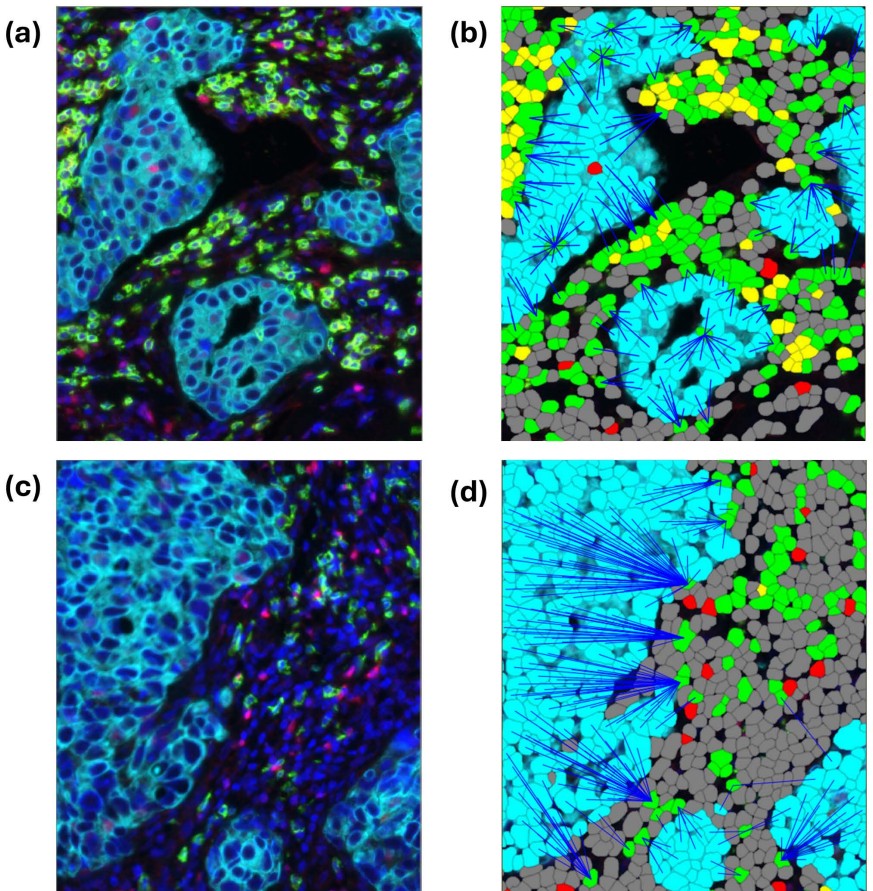

**Fig 2. Multiplex immunofluorescent (IF) image of breast cancer tissue and corresponding cell segmentation.** Multiplex IF image of breast cancer tissue. (**a** and **c**) Cells are stained with 4 colors indicating biomarkers for Hoechst (blue), Cytokeratin (cyan), CD8 (green), and FoxP3 (red). (**b** and **d**) Multiplex IF image with overlayed colored tiles indicating cell classification: Epithelial (cyan), CD8+T cell (green), CD8+/FoxP3+ (yellow), FoxP3 only (red), and unclassified (dark gray). Cells are detected using the Hoechst channel and classified as epithelial, lymphocytes, or unclassified based on the intensity of the Cytokeratin, CD8, and FoxP3 channels. Distance is computed from each epithelial cell to the nearest CD8+T cell (length of blue line). The distances are used to compute proximity and consistency. Image (a) represents a high-proximity specimen, while (c) represents a specimen with lower proximity.

Finally, we considered the levels of correlation between all computed spatial metrics (S3 Table). Proximity and count showed moderate levels of correlation (corr. = 0.5545). This was not unexpected, given that cell distances are inherently influenced by cell density. However, the correlation between count and consistency was low (corr. = 0.3261). Furthermore, proximity and consistency had a relatively low correlation (corr. = 0.4717).

### Recurrence-free survival analysis

Continuous proximity, consistency, and count scores were included in Cox proportional hazard models to assess the significance of the spatial measurements (Table 1). For interpretability, hazard ratios were computed such that higher values of continuous variables (the hypothesized good prognosis phenotype) was the referent. Both adjusted and unadjusted hazard ratios were presented, with each spatial score adjusted for the other two variables. Interaction between proximity and count was also considered due to the moderate correlation between the two variables. In univariate models, all spatial variables showed evidence of association with recurrence, with the strongest evidence for proximity ($p<0.001$) and

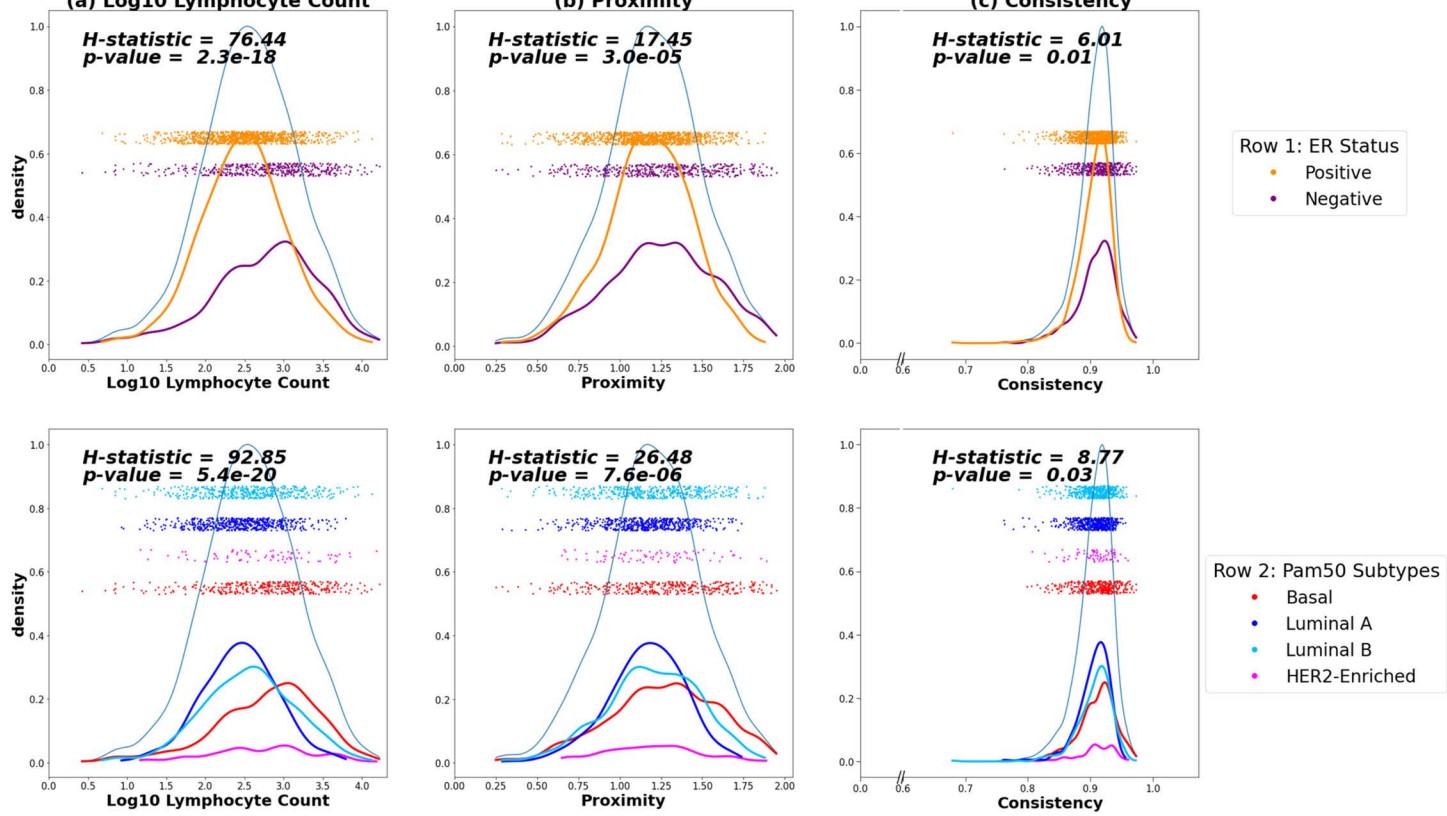

**Fig 3. Distribution of count, proximity, and consistency by estrogen receptor (ER) status and PAM50 subtypes.** Density plots for the **(a)** log base 10 lymphocyte counts, **(b)** proximity, and **(c)** consistency values in Phase 3 Carolina Breast Cancer Study (CBCS) participants with sub-densities and overlayed with jittered scatter plot of the data colored by ER status (top row) and PAM50 gene signature subtypes (bottom row). *H*-statistics and *p*-values correspond to the Kruskal–Wallis test, a nonparametric version of the analysis of variance test.

**Table 1. Cox proportional hazards model for continuous spatial metrics.**

| Covariates | Multivariate HR (95% CI) | Multivariate *p*-value | Univariate HR (95% CI) | Univariate *p*-value |
|---|---|---|---|---|
| **Proximity** | 2.04 [1.12, 3.70] | 0.0189 | 2.46 [1.54, 3.92] | <0.001 |
| **Consistency** | 1.14 [1.00, 1.30] | 0.0554 | 1.23 [1.08, 1.37] | <0.001 |
| **Count** | 2.24 [0.65, 7.67] | 0.1989 | 1.24 [1.02, 1.51] | 0.0341 |
| **Proximity × Count** | 1.62 [0.78, 3.36] | 0.1965 | NA | NA |

The model controls for count and potential count interactions.

95% CI, 95% Confidence Interval; HR, Hazard Ratio; NA, Not Applicable.

consistency (*p* < 0.001). Proximity showed the strongest association with RFS across all samples (HR: 2.46, 95% CI [1.54, 3.92]; *p* < 0.001). In multivariate models adjusted for other spatial factors, proximity maintained a strong effect size with a hazard ratio of 2.04 (95% CI [1.12, 3.70]; *p* = 0.0189). Consistency showed some evidence of association with recurrence and had a hazard ratio of 1.14 (95% CI [1.00, 1.30]; *p* = 0.0554) for RFS overall. There was no evidence to suggest that count or the interaction between count and proximity were associated with recurrence after adjusting for other spatial metrics.

Proximity and consistency were included in a Cox proportional hazards model with important clinical and demographic variables (Table 2). In multivariable models adjusted for age, stage, grade, ER status, PR status, and HER2 status, proximity continued to show strong evidence of association with a hazard ratio of 2.88 (95% CI [1.70, 4.90]; $p < 0.0001$). However, there was no evidence that consistency was associated with recurrence after adjusting for clinical variables and age ($p = 0.2060$).

Kaplan–Meier estimates of the RFS functions for dichotomized spatial metrics are shown in Fig 4, with hazard ratios for the univariate and multivariate Cox proportional hazards model in Fig 5. Consistent with a priori knowledge, ER-negative tumors showed associations between lymphocyte counts and improved outcomes (Fig 4, third row, $p = 0.027$). Higher lymphocyte count was associated with longer survivorship (HR 1.61; 95% CI [1.05, 2.45]; $p = 0.0284$ for low versus high counts). However, proximity also showed associations, with comparable HRs (HR 1.84; 95% CI [1.18, 2.87]; $p = 0.0069$). Consistency showed little evidence of association with recurrence ($p = 0.0576$).

Associations for count and proximity diverged when considering ER+ and ER− tumors together. Among breast cancers overall, high proximity and high consistency both showed strong evidence of association with recurrence ($p < 0.001$), while there was little evidence that lymphocyte count was associated with recurrence (Fig 4, top row). The hazard ratio for proximity was 1.94 (95% CI [1.46, 2.58]; $p < 0.0001$ low versus high proximity), and low consistency participants had higher recurrence, with a hazard ratio of 1.62 (95% CI [1.22, 2.15]; $p = 0.0008$ low versus high consistency). Lymphocyte count had the weakest association overall, with a hazard ratio of 1.29 (95% CI [0.98, 1.71]; $p = 0.0709$).

Among ER-positive participants, high proximity and consistency were strongly associated with lower recurrence (Fig 4, middle row, $p$-values $< 0.01$). The hazard ratios for low versus high proximity and consistency are 2.04 (95% CI [1.39, 2.98]; $p = 0.0003$) and 1.82 (95% CI [1.23, 2.69]; $p = 0.0026$), respectively, similar to those for cancers overall. In contrast, the hazard ratio for low versus high lymphocyte count was 1.35 (95% CI [0.92, 1.98]; $p = 0.1289$) (Fig 5). The results of the RFS analysis are similar when including CD8+/FoxP3+ cells to compute proximity and consistency (S2 Fig). This suggests that while lymphocyte count is not associated with recurrence in ER-positive participants, spatial characteristics described by proximity and consistency are frequently associated with recurrence irrespective of ER status.

To assess the independent value of proximity and consistency relative to models with lymphocyte count alone, we used nested Cox proportional hazards models and Likelihood Ratio Tests (S4 Table). We found that including proximity and consistency improved the fit over the model with count alone ($p < 0.0001$). Furthermore, when adjusting for count, hazard ratios for proximity and consistency did not differ significantly from the unadjusted hazard ratios, overall and among ER-positives (Fig 5).

**Table 2. Cox proportional hazards model for continuous spatial metrics adjusting for clinical variables.**

| Covariates | Multivariate HR (95% CI) | Multivariate *p*-value |
|---|---|---|
| Proximity | 2.88 [1.70, 4.90] | <0.0001 |
| Consistency | 1.10 [0.95, 1.26] | 0.206 |
| Age | 1.01 [0.99, 1.02] | 0.3397 |
| Grade | 1.43 [1.11, 1.83] | 0.0051 |
| Stage | 1.79 [1.49, 2.13] | <0.0001 |
| ER-Negative | 1.29 [0.83, 2.01] | 0.2655 |
| PR Negative | 1.10 [0.71, 1.71] | 0.6588 |
| HER2 Negative | 1.38 [0.90, 2.10] | 0.1375 |

The model controls for Age, Grade, Stage, Estrogen Receptor (ER) status, Progesterone Receptor (PR) status, and Human Epidermal Growth Factor Receptor 2 (HER2) status.

95% CI, 95% Confidence Interval; HR, Hazard Ratio.

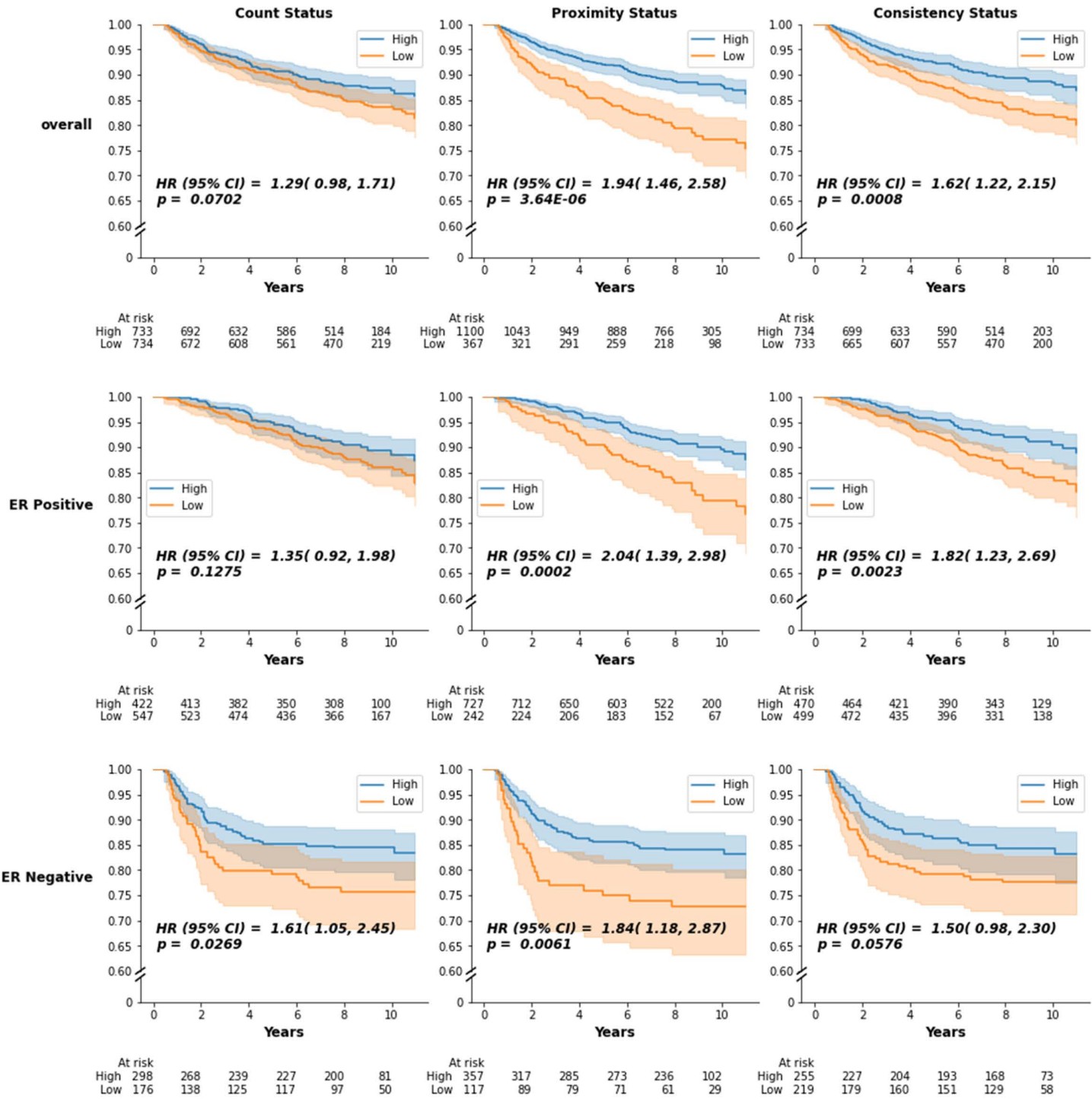

**Fig 4. Kaplan–Meier survival curves for count, proximity, and consistency overall participants and participants stratified by estrogen receptor (ER) status.** Kaplan–Meier estimates for recurrence-free survival (RFS) comparing high vs. low proximity, consistency, and lymphocyte count in Phase 3 CBCS participants overall and stratified by ER status. 95% confidence bounds are given by the shaded region. Table of number of at risk participants is given below each plot. *P*-values (*p*) correspond to the log-rank test of the difference between curves. Hazard Ratios (HR) and 95% Confidence Intervals (95% CI) are given for the difference between the hazards of the survival curves of the high vs. low status.

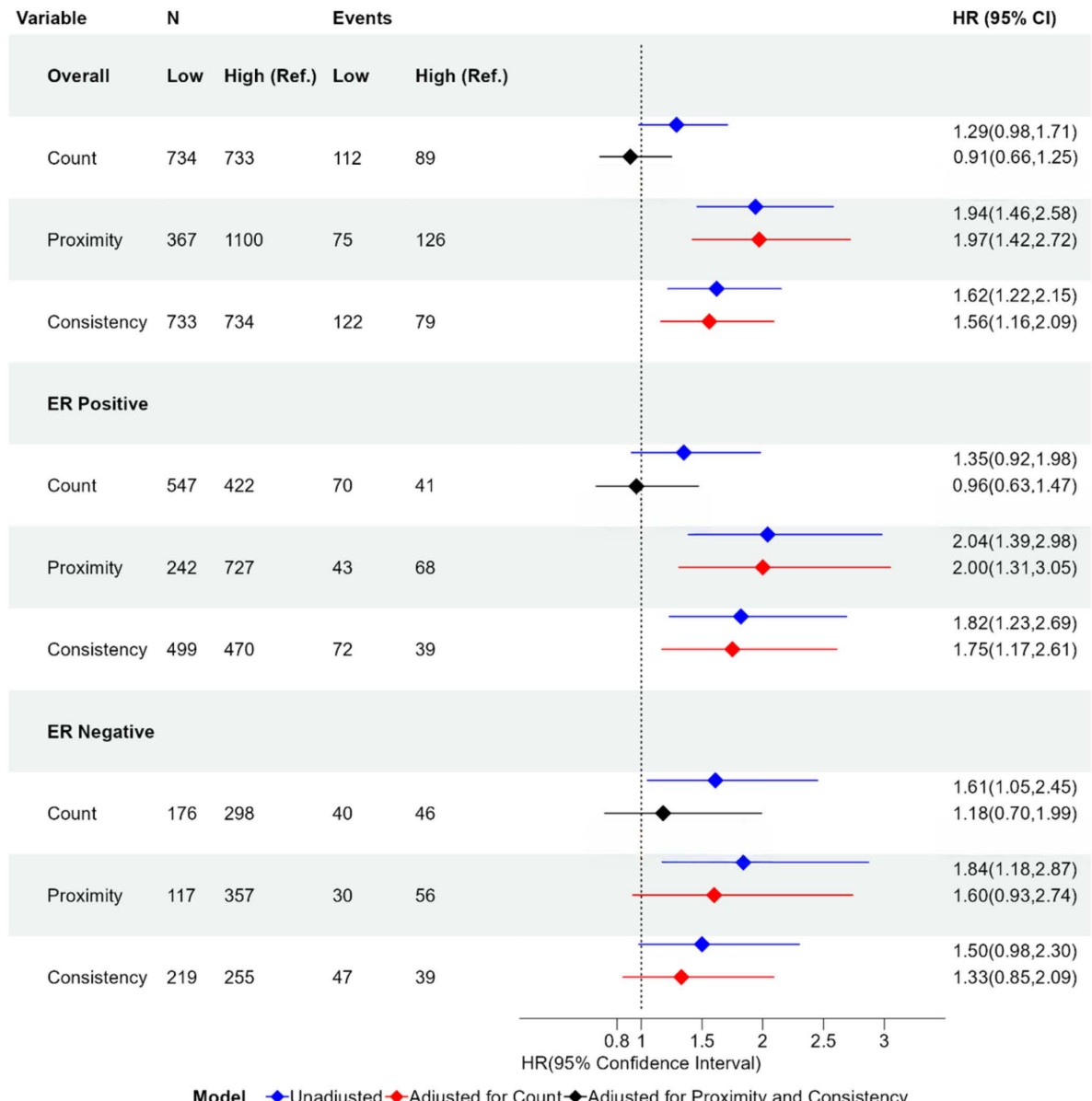

**Fig 5. Forest plot of hazard ratios for count, proximity, and consistency overall participants and stratified by estrogen receptor (ER) status.** Forest plot of hazard ratios and 95% confidence intervals for proximity, consistency, and lymphocyte count binary variables comparing low and high values when considering all participants and stratified by ER status. The hazard ratio is given in the unadjusted case (blue), adjusted for lymphocyte count (red), and adjusted for both proximity and consistency (black). The hazard ratio is with respect to the reference group (Ref.). The reference groups are High Count, High Proximity, and High Consistency. Group size (*N*) and number of participants who experience recurrence (Event) are given for each group. Hazard Ratio (HR); 95% Confidence Interval (95% CI).

Similarly, we assessed whether lymphocyte count provided independent value in models that include proximity and consistency. Compared to a model that included proximity and consistency only, LRT tests showed that including count in the model did not improve the fit over the model that only included proximity and consistency (*p* = 0.55). Moreover, when adjusting the effects of count by including proximity and consistency, the hazard ratio decreased significantly (Fig 5). Even

among ER-negatives, the hazard ratio for count after adjusting for proximity and consistency showed no evidence of association with recurrence (1.18, 95% CI [0.70,1.99]; $p = 0.53$).

Given that count and proximity were moderately correlated, we also considered stratified models to assess the relationship between count and proximity (Fig 6). Among participants with low lymphocyte counts, high proximity was associated with improved outcomes ($p = 0.0002$). Among participants with high lymphocyte counts, high consistency remained associated with improved survival ($p = 0.0005$). Thus, proximity and consistency have independent value beyond lymphocyte count in predicting recurrence.

## Comparison of tissue microarray-based spatial assessment of TILs to whole slide TIL analyses

To contextualize our measures of immune spatial relationships relative to established biological metrics, we compared the proximity and consistency measurements (determined based on tissue microarrays from the CBCS) to a set of spatial TIL scores from the slides of the same participants. In the slides, larger-scale metrics, including TILs forests, TILs deserts, intratumoral strength, and peritumoral strength, were estimated as described by Fassler [25]. Notably, proximity and consistency can be determined in 1 mm diameter tissue microarray cores, whereas the whole slide images evaluated for forests and deserts consist of tissue samples which are 15–25 mm across the longest dimension. On this larger scale, TIL forest and desert scores address presence or absence of TILs, respectively, and intratumoral strength and peritumoral strength describe the level of TILs in the tumor interior and periphery, respectively.

Proximity, consistency, and lymphocyte count were all correlated with the four whole slide image TIL scores (Fig 7). We observed the strongest correlations between proximity and intratumoral strength (corr. = 0.60), but also observed associations of proximity with peritumoral strength (corr. = 0.57) and TIL forests (corr. = 0.50) and an inverse association with TIL deserts (corr. = −0.47) (Fig 7). Consistency had a weaker correlation due to a parabolic relation with intratumoral and peritumoral strength. As scores increased beyond 1, consistency increases. Still, consistency was correlated with TIL forests (corr. = 0.32) and inversely with TIL deserts (corr. = −0.36) scores, but more weakly than proximity and lymphocyte count. Taken together, these analyses suggest that our small-scale spatial metrics from tissue microarray cores reflect prognostically relevant, large-scale TIL spatial structures. We performed sensitivity analyses adjusting for whole slide image TIL scores, and while these analyses were underpowered due to missing whole slide image data, we observed similar magnitude associations (S5 Table).

## Relation to gene expression biomarkers

We next hypothesized that spatial parameters for immune infiltration may predict molecular features as measured in RNA. We queried both immune signatures and signatures of genomic instability. Hamilton and colleagues 2022 identified three distinct immune classes (AE, Innate-Enriched (IE), or Immune Quiet) based on gene expression that were associated with TIL counts [24]. As shown in Table 3, AE tumors were associated with high proximity, consistency and lymphocyte count compared to the combined IE and Immune Quiet (referent) with odds ratios of 4.5 (95% CI [3.0, 7.1]; $p < 0.0001$), 2.4 (95% CI [1.8, 3.2]; $p < 0.0001$), and odds ratio of 6.0 (95% CI [4.4, 8.4]; $p < 0.0001$) respectively. Additionally, a biomarker of AGI biomarker (defined by high homologous recombination deficiency (HRD) and/or TP53 Mutant-like signatures; versus NGI markers), was associated with Adaptive immune class in previous work [12], and was associated with high lymphocyte count (OR 2.5; 95% CI [1.9, 3.3]; $p < 0.0001$) but was not strongly associated with proximity and consistency. In sensitivity analysis considering RNA-based immune subtypes, hazard ratios for proximity and consistency were largely unchanged after adjusting for RNA immune subtypes (S6 Table and S7 Table).

## Discussion

While the clinical significance of immune response in breast cancers is established, uncertainty remains about the importance in non-triple negative subtypes (i.e., ER-positives). The best metrics for immune-related prognosis are still debated. Our work explored spatial parameters of lymphocytes in tumor and peritumoral regions as predictors of recurrence.

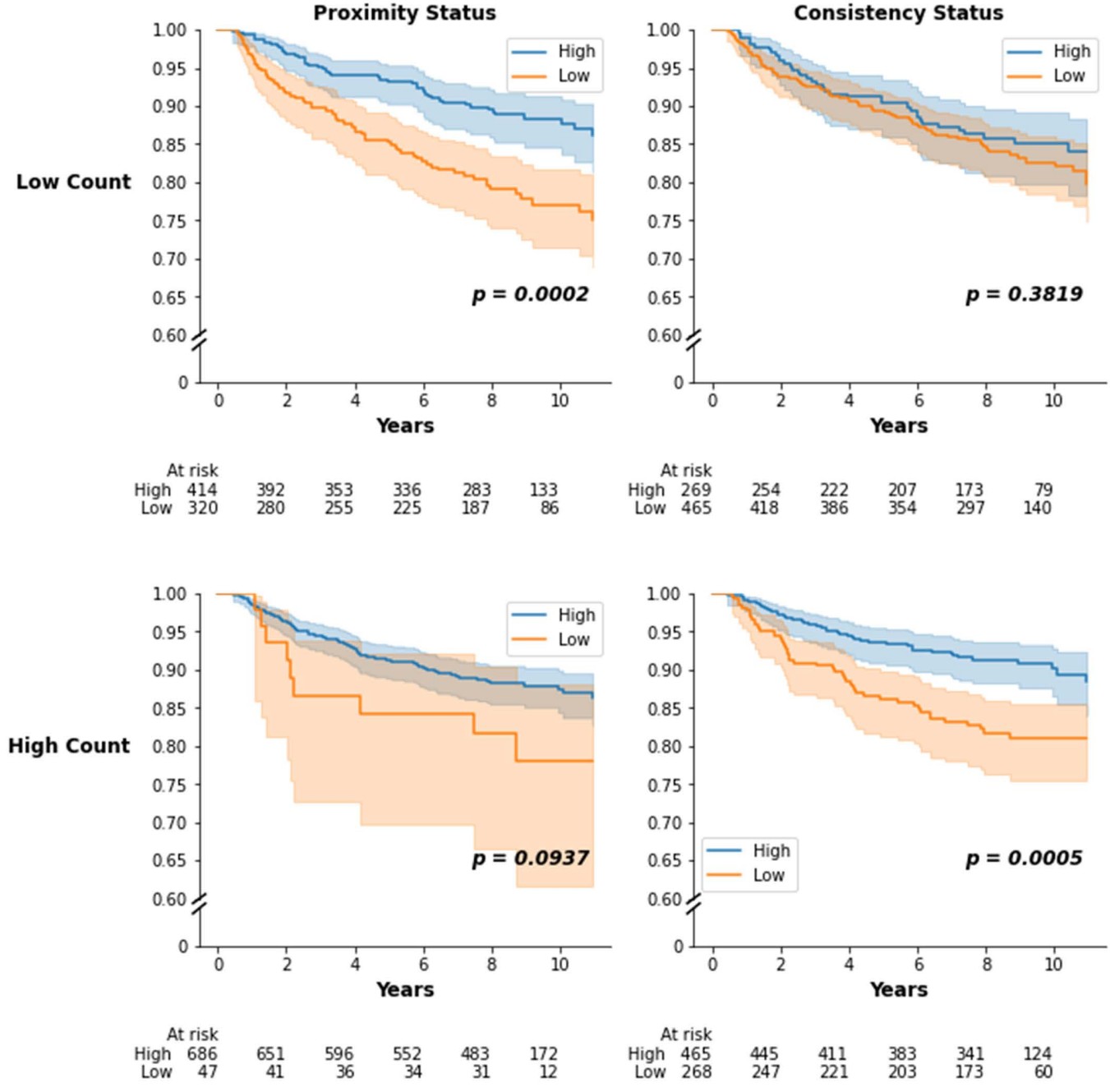

**Fig 6. Kaplan–Meier survival curves for proximity and consistency stratified by high and low lymphocyte count.** Kaplan–Meier estimates for recurrence-free survival (RFS) comparing high vs. low proximity and consistency in Phase 3 Carolina Breast Cancer Study (CBCS) participants stratified by high vs. low lymphocyte count, defined as above the median count or below the median count. 95% confidence bounds are given by the shaded region. *P*-values (*p*) correspond to the log-rank test of the difference between curves. The difference in RFS curves for high vs. low proximity is more significant in participants with low lymphocyte count. The difference in RFS curves for high vs. low consistency is more significant in participants with high lymphocyte count.

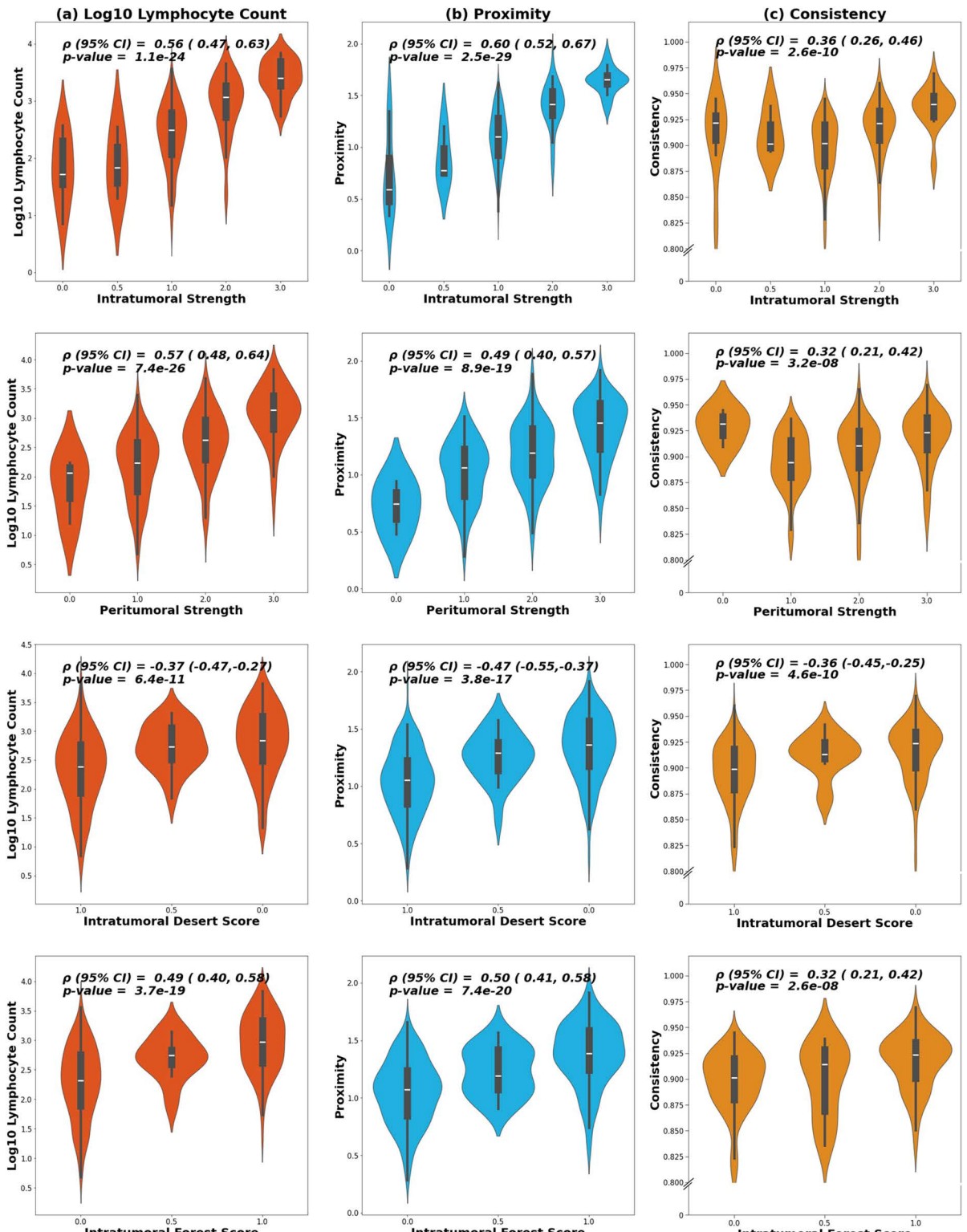

**Fig 7. Violin plots of distribution of proximity, consistency, and count for Whole Slide Image tumor infiltrating lymphocyte (TIL) scores.** Violin plots depicting the distribution of **(a)** proximity, **(b)** consistency, and **(c)** log10 count for the scores of intratumoral strength, peritumoral strength, TIL Deserts, and TIL Forests given in the paper by Fassler [25]. Spearman's correlation coefficient ($\rho$) and $p$-values for coefficient significance are given. Note

that only 288 of the total participants were scored for TIL Forests, Deserts, intratumoral strength, and peritumoral strength. 95% Confidence Intervals (95% CI) for Spearman's correlation coefficient are computed using Fisher's z-transformation.

**Table 3. Contingency tables for spatial biomarkers vs. gene expression-derived immune classes.**

| | Proximity | | | | Consistency | | | | Lymphocyte Count | | | |
|---|---|---|---|---|---|---|---|---|---|---|---|---|
| | High | Low | Total | Odds Ratio (95% CI) | High | Low | Total | Odds Ratio (95% CI) | High | Low | Total | Odds Ratio (95% CI) |
| AE[a] | 281 | 31 | 312 | 4.5 [3.0, 7.1] | 199 | 113 | 312 | 2.4 [1.8, 3.2] | 238 | 74 | 312 | 6.0 [4.4, 8.4] |
| IE/Quiet[b] | 401 | 201 | 602 | | 256 | 346 | 602 | | 209 | 393 | 602 | |
| Total | 682 | 232 | 914 | | 455 | 459 | 914 | | 447 | 467 | 914 | |
| AGI[c] | 331 | 110 | 441 | 1.1 [0.8, 1.5] | 234 | 207 | 441 | 1.3 [1, 1.7] | 264 | 177 | 441 | 2.5 [1.9, 3.3] |
| NGI[d] | 308 | 115 | 423 | | 196 | 227 | 423 | | 159 | 264 | 423 | |
| Total | 639 | 225 | 864 | | 430 | 434 | 864 | | 423 | 441 | 864 | |

95% Confidence Interval (95% CI).

[a]Adaptive-Enriched (AE) from Hamilton and colleagues (2022) [24].

[b]Innate-Enriched (IE) and Immune Quiet (Quiet) from Hamilton and colleagues (2022) [24].

[c]Any Genomic Instability (AGI) from Hamilton and colleagues (2023) [12].

[d]No Genomic Instability (NGI) from Hamilton and colleagues (2023) [12].

Proximity and consistency of lymphocytes were significant determinants of recurrence for both ER+ and ER− breast cancers, while lymphocyte count showed weaker associations, especially among ER+. Lymphocyte count did not offer independent value in prognostic models with proximity and consistency. A possible explanation for the relative value of proximity in ER+ relates to an inherent relation between count and proximity; when counts are high, proximity is also de facto high. However, when lymphocyte count was low, spatial proximity was important and associated with improved recurrence-free survival. These data suggest that a lower concentration, well-targeted immune response, has efficacy in slowing progression. Furthermore, consistency measures the level of uniformity of CD8+T cells among tumor cells as well as the inter-core heterogeneity of immune response. A more uniform immune response may imply greater efficiency in the cytotoxic effects of CD8+T cells. Additionally, proximity and consistency were both correlated with other spatial TILs scores drawn from whole slide images. Both were also associated with RNA-based AE gene expression. These results imply that the location of CD8+T cells in relation to tumor cells have prognostic and biological significance and that sensitive spatial parameters are important for characterizing the tumor immune microenvironment.

Our work is consistent with several previous studies demonstrating that higher levels of CD8+T cells are associated with positive prognosis in TNBC [29,30]. Most of these studies have suggested limited or no benefit of CD8+T cell infiltration for ER+ participants [31,32]. However, one prior study documented CD8+T cells in stroma and intratumoral regions and established an association between intratumoral T cells and survival for both ER+/HER2− and ER− participants [33] Another study used QuPath, the same open-source digital pathology software used herein, to build a machine learning algorithm for scoring HE images based on density of TILs in various tissue compartments [34] and confirmed prognostic significance of TILs in TNBC, but did not evaluate outcomes in ER+ or ER−/HER2+ breast cancer [34] None of these studies focused on spatial proximity and consistency, reflecting the general tendency of the prior literature to emphasize CD8+T cell counts.

Two recent studies considered spatial parameters of CD8+T cells [13,14]. One study established that density of CD8+T cells within 20 µm of breast cancer tumors is predictive of pathologically complete response and disease-free survival in neoadjuvant therapy [14]. However, their methods still emphasized cell density (albeit within a defined range) as opposed

to using distance to measure spatial differences. Moreover, they did not consider survival differences when stratifying by cancer subtypes. The other study considered proximity of CD8+T cells to pancreatic ductal adenocarcinoma cells and observed longer overall survival when CD8+T cells were closer to focal adhesion kinase (FAK)+ tumor cells [13]. Inconsistencies in associations with ER+ disease in the previous literature may reflect unmeasured parameters or measurement of a factor that is correlated with an unknown metric; our data suggests that proximity and consistency are two plausible underlying parameters. For CD8+T cells to induce cytotoxic response, they must be in direct contact with tumor cells [20], and therefore distance-based methods have theoretical advantages for naturally quantifying interactions between CD8+T cells and tumor cells.

Our work should also be considered in context of methods of automatic spatial characterization of TILs in HE images [25,34–36] Abousamra and colleagues used convolutional neural networks to create whole slide image TILs maps by classifying tiled image patches as TIL positive or negative (based on at least 2 lymphocytes detected) [36] These models have advantages in using complex deep learning models. However, classifying image tiles could miss fine-grained variation in the tumor immune microenvironment and lead to misclassification of proximity. Our use of simple distance-based methods eases scalability to whole slide images and is well-suited to tissue microarrays that are commonly available for large cohorts. Moreover, proximity and consistency of lymphocyte infiltration at the tissue microarray level were correlated with measures of TIL intratumoral strength and peritumoral strength at the whole-slide level. This suggests that spatial metrics applied to tissue microarray are predictive of large-scale spatial immune structures within the tumor.

A strength of our analysis was the ability to compare the spatial metrics from IF images to bulk gene expression data from the same specimens. The connection between gene expression-derived immune clusters [24] and the spatial biomarkers suggests that CD8+T cells function in the tumor immune microenvironment, reflect broader patterns of immune cells. However, we were unable to fully interrogate the relationship between multiple immune markers and spatial arrangement. Proximity and consistency could be affected by interactions with other lymphocyte types, with some previous work suggesting that increased regulatory T cells could impair CD8+T cell activity. Furthermore, we observed that genomic instability based on TP53 and HRD [12] did not predict proximity and consistency of immune cells, whereas lymphocyte count was associated with genomic instability. This may reflect the relationship between neo-antigen production and a very robust immune response. However, we lack data to specifically interrogate neoantigens as a mediator of count versus spatial immunogenic patterns. A limitation of our analysis is that we did not validate our IF measures across the larger area of whole slides, however, we find it promising that proximity and consistency were associated with whole slide immune features. Finally, while IF allowed us to distinguish lymphocyte types, we studied CD8+T cells and neglected other spatial interactions between immune cell types. However, CD8 is a priority marker given its established role in cytotoxic immune responses.

In summary, our analysis suggests prognostic value in the location of lymphocytes, specifically CD8+T cells. If validated, proximity and consistency may identify ER-positive breast cancers that benefit from immune therapy, expanding treatment options to include as many as 75% of ER+ tumors (i.e., the proportion of cancers in this dataset that showed high proximity). Multiplex IF staining is increasingly cost-effective and efficient, allowing for viable implementation in clinical settings [18]. Therefore, future clinical studies may be able to use proximity and consistency of multiplex IF-derived spatial immune markers to predict response to immunotherapy. It is also possible that HE-derived distance metrics, if validated, could be useful. Given that ER+ tumors represent the majority of breast tumors, closer evaluation of immune response to identify possible beneficiaries is highly impactful.

## Supporting information

**S1 Fig. Quality control flow chart.** Diagram shows how many participants were excluded from the analysis due to quality control failures. Furthermore, the number of participants included in various analyses and with available WSI image and gene expression-derived classifications is presented.
(PNG)

**S2 Fig. Forest plot of hazard ratios for count, proximity, and consistency overall participants and stratified by ER status when including double positive cells to compute proximity and consistency.** Hazard Ratios for Relapse Free Survival combining CD8+/FoxP3− and CD8+/FoxP3+ lymphocytes to compute proximity and consistency. The hazard ratios (diamond) and 95% confidence intervals (lines) are shown for proximity, consistency, and lymphocyte count binarized at the median (high as referent), overall, and stratified by ER status. Estimates were unadjusted (blue), adjusted for lymphocyte count (red), and adjusted for both proximity and consistency (black) where relevant. The referent categories (Ref.) were defined as High Count, High Proximity, and High Consistency. Group size (*N*) and number of participants who experience recurrence (Event) are given for each group. HR: Hazard Ratio; 95% CI: 95% confidence interval.
(PNG)

**S1 Table. Tuning parameters for cell detection and classification.** Table of tuning parameters for cell detection and thresholds for classification of cells used in QuPath functions.
(DOCX)

**S2 Table. Sensitivity analyses to optimize cut points for count, proximity, and consistency training and test sets.** The results from optimizing the cut point for the high versus low count, proximity, and consistency. We split the data based on a 2/3 to 1/3 training and testing split. Furthermore, we used stratified sampling on the recurrence data to ensure there would be an adequate number of participants with recurrence in the training and testing sets. We find an optimal cut point based on the quantile of the data, which minimizes the *p*-value for the log-rank test. Though the cut points appear to perform well on the training set, in most cases, there appears to be overfitting based on very different hazard ratios in the testing data. Furthermore, many results which were statistically significant in the training data are no longer statistically significant in the testing data.
(DOCX)

**S3 Table. Correlation matrix for spatial metrics.**
(DOCX)

**S4 Table. Summary of likelihood ratio tests for nested multivariate Cox proportional hazards models.** A summary of multivariate Cox proportional hazards models with proximity, consistency, and lymphocyte count binary covariates for all participants. Estimates for the covariate-adjusted hazard ratios are provided, as well as *p*-values for the significance of the hazard ratios. Models with nested covariates are compared using the likelihood ratio test, and *p*-values for the significance of the difference between compared nested models are provided. The reference groups for each of the covariates are high proximity, high consistency, and high lymphocyte count, respectively.
(DOCX)

**S5 Table. Cox proportional hazards model with WSI TIL scores.**
(DOCX)

**S6 Table. Cox proportional hazards model with RNA-based immune classes.**
(DOCX)

**S7 Table. Cox proportional hazards model with CD8 T cell signature.**
(DOCX)

**S1 Checklist. REMARK checklist.** Checklist of reporting guidelines for biomarkers [21] providing section and paragraph number(s) where each item is addressed.
(DOCX)

## Acknowledgments

The authors would like to acknowledge the University of North Carolina BioSpecimen Processing Facility for sample processing, storage, and sample disbursements (http://bsp.web.unc.edu/). We are grateful to CBCS participants and study staff. The findings and conclusions in this publication are those of the authors and do not necessarily represent the views of the North Carolina Department of Health and Human Services, Division of Public Health. The views and conclusions contained in this document are those of the authors and should not be interpreted as representing the official policies, either expressed or implied, of the U.S. Government.

## Author contributions

**Conceptualization:** Andrew E. Walker, Didong Li, James Steve Marron, Katherine A. Hoadley, Melissa A. Troester.

**Data curation:** Xiaohua Gao, Qichen Wang, Gabriela De la Cruz, Joel Saltz.

**Formal analysis:** Andrew E. Walker, Qichen Wang.

**Funding acquisition:** Charles M. Perou, James Steve Marron, Katherine A. Hoadley, Melissa A. Troester.

**Investigation:** Andrew E. Walker, Xiaohua Gao, Melissa A. Troester.

**Methodology:** Andrew E. Walker, James Steve Marron.

**Project administration:** Katherine A. Hoadley, Melissa A. Troester.

**Resources:** Xiaohua Gao.

**Supervision:** Didong Li, James Steve Marron, Katherine A. Hoadley, Melissa A. Troester.

**Validation:** Charles M. Perou.

**Visualization:** Andrew E. Walker.

**Writing – original draft:** Andrew E. Walker, Qichen Wang.

**Writing – review & editing:** Xiaohua Gao, Gabriela De la Cruz, Didong Li, Charles M. Perou, Joel Saltz, James Steve Marron, Katherine A. Hoadley, Melissa A. Troester.

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
