## [Editor Report · Decision Letter 0]

23 May 2025

Dear Dr Walker, 

Thank you for submitting your manuscript entitled "Prognostic Significance of CD8 T-cell Spatial Biomarkers in ER+ and ER- Breast Cancer" for consideration by PLOS Medicine.

Your manuscript has now been evaluated by the PLOS Medicine editorial staff and I am writing to let you know that we would like to send your submission out for external peer review.

For clinical studies, please upload a copy of your trial study protocol as a supporting information file. The study protocol should be the version submitted for approval to the institutional review board or ethics committee, should include any amendments to the study protocol, as well as the date of their approval by the institutional review or ethics committee. Please also detail any deviations from the study protocol in the Methods section of your manuscript. The editors will consider the protocol and study conduct prior to a final decision for external review. 

Please re-submit your manuscript within two working days, i.e. by May 27 2025.

Feel free to email me at atosun@plos.org or us at plosmedicine@plos.org if you have any queries relating to your submission.

Kind regards,

Alexandra Tosun, PhD

Associate Editor

PLOS Medicine

---

## [Decision Letter · Decision Letter 1]

9 Jul 2025

Dear Dr Walker,

Many thanks for submitting your manuscript "Prognostic Significance of CD8 T-cell Spatial Biomarkers in ER+ and ER- Breast Cancer" (PMEDICINE-D-25-01829R1) to PLOS Medicine. The paper has been reviewed by subject experts and a statistician; their comments are included below and can also be accessed here: [LINK]

As you will see, the reviewers describe the study as highly interesting and timely and provide suggestions to enhance clarity, rigor, and impact. After discussing the paper with the editorial team, I'm pleased to invite you to revise the paper in response to the reviewers' comments. We plan to send the revised paper to some or all of the original reviewers, and we cannot provide any guarantees at this stage regarding publication.

We ask that you submit your revision by 30 July 2025. However, if this deadline is not feasible, please contact me by email, and we can discuss a suitable alternative.

Don't hesitate to contact me directly with any questions (atosun@plos.org). 

Best regards, 

Alexandra 

Alexandra Tosun, PhD 

Senior Editor

PLOS Medicine

atosun@plos.org

Comments from the reviewers: 

Reviewer #1: This research investigated the prognostic value of two novel spatial biomarkers in various types of breast cancer. Specifically, the biomarkers represented the proximity and consistency of immune cells with respect to cancer cells. The novel biomarkers demonstrated prognostic value in both ER+ and ER- breast cancers and were consistent with other established tumour immune biomarkers. The findings suggest the novel biomarkers may have predictive utility in identifying more individuals who would benefit from immune therapy than standard of care. 

The manuscript is an excellent initial submission and I commend the authors for establishing the importance of their research questions and clearly communicating complex laboratory methods. I do have some statistical and methodological concerns I have discussed in detail below. Most notably, I would encourage the authors to focus more on the effect size and clinical importance of their findings, instead of p-values. I'd also encourage the authors to make use of the REMARK guidelines, including explanations and elaborations, both as a tick-the-box exercise to ensure everything is covered and to pressure test the validity of their research (https://doi.org/10.1371/journal.pmed.1001216). 

Detailed feedback on the manuscript is provided below. My primary focus has been methodological, but I have made a few broader points as well. All items are major unless indicated otherwise.

1. [MINOR] Please double check that all abbreviations are defined the first time they are used e.g., "IF" in the Abstract, "ER" in the Introduction, "TMA" used in Study Population before being defined in TMA Staining and Processing.

2. [MINOR] Consider incorporating very brief definitions of "proximity" and "consistency" in the abstract. I understand word count makes this tricky, but these are the main innovation being presented. 

3. The authors say "Very few studies have evaluated proximity or other spatial characteristics and few have used specific cell-type markers in analyses… such as those available via multiplex immunofluorescent." but don't yet convincingly demonstrate this is the case. A quick search indicates there are more recent publications on this topic than those cited by the authors. Suggest more clearly establishing the gap in the existing literature by citing and succinctly critiquing the few available study examples and, if possible, citing relevant reviews that support the need for their research. 

4. [MINOR] In the Introduction the authors talk generally about "mean distance" between tumour and immune cells, however they clarify later it's specifically nearest neighbour. Suggest tweaking wording to make this clear from the start. 

5. Given the cascade of participant exclusions presented under "Study Population" and the varying samples for various analyses, consider including a CONSORT-style figure that represents this visually.

6. I appreciate the care and clarity taken by the authors when presenting lab methods. They are readily comprehensible, even for a statistician! 

7. [MINOR] The authors chose to consider a cell that exceeds the thresholds for both CK and CD8 or FoxP3 as a lymphocyte because "A cell cannot be both a tumor cell and a lymphocyte…". What was the reason for this decision? Note I am not suggesting this is incorrect - this is well outside my expertise - just that I am interested in the reasoning. The authors could also consider including a reference to support their decision. 

8. With respect to dichotomising various continuous variables (e.g., at the median value), while this historically has been common practice in translational and biomarker research, recent methodological guidance discourages this (refer the REMARK guideline paper linked above as well as https://doi.org/10.1136/bmj.332.7549.1080). I suggest the authors consider modelling continuous predictors and outcomes as continuous variables, including more contemporary techniques like restricted cubic splines to address non-linearity if encountered. 

9. Relatedly and more seriously, using model results like hazard ratios to retrospectively merge quartiles is a practice at high risk of bias (see earlier links plus this more methodological resource https://doi.org/10.1002/sim.2331). I encourage the authors to reconsider this approach. 

10. When presenting results from a Cox regression, if a p-value is to be presented in addition the to the HR and CI then the p-value should come from the Cox model, not a log-rank test on the same data. The log-rank test can be used in a simpler, unadjusted analyses but mixing a log-rank test p-value with Cox regression results is confusing at best and misleading at worst. 

11. [MINOR] Relatedly, while hazard ratios are the generally accepted statistic for time-to-event analyses, they have some well-recognised limitations (see https://doi.org/10.1093/ehjacc/zuae017 and https://evidence.nejm.org/doi/10.1056/EVIDe2300142). The authors may wish to consider supplementing key HR outcomes with another statistic like difference in median survival or restricted mean survival time to assist with interpretability.

12. While the use of Spearman's rho as described is valid, it is limited in that it only provides an estimate of the association, not the effect size. Suggest the authors consider linear regression (or similar) either instead of or in addition to Spearman's rho, noting care must be taken when modelling a non-linear relationship. 

13. The authors make many different references to "(statistical) significance". This is risky because the more claims of significance are made, the greater the risk of false positives and the more important it becomes to consider multiple testing correction strategies like the Bonferroni adjustment. Assuming the authors would prefer to avoid this, I'd suggest moderating or changing language away from statistical significance as an all-or-nothing threshold. One option is to talk in terms of strength of evidence e.g., "strong/moderate/some evidence of…" or "no evidence was found of…". I suggest avoiding statements like "borderline significantly" (particularly when the p > 0.05) and "… most significant association…" (for variable with smallest p-value in the model) as these can overinterpret and potentially mislead. 

14. There are many methods discussed for the first time in the Results. Suggest ensuring the demarcation between Methods and Results is carefully observed. 

15. Similarly, suggest ensuring any non-trivial interpretation of Results is presented in the Discussion, not the Results. 

16. [MINOR] Avoid using exponential notation - for p-values suggest something like "p<.0001" (or similar) for very small values. 

17. I found Figure 4 quite difficult to understand - perhaps revisit and ensure spacing and labelling are correct. 

18. I understand and agree with the authors' use of the LRT to assess the joint contribution of proximity and consistency to the Cox proportional hazards model. That said, care must be taken when interpreting the results. A small p-value indicates that the variables improve model fit but does not necessarily mean this improvement is clinically meaningful or that the added variables provide substantial predictive value.

19. I have already noted that much of the Results are actually methods that should be moved to the Methods section. I would also suggest more detail be added to the Methods about the modelling underpinning the analyses in Table 1, because currently odds ratios are presented without clearly specifying the generating model. 

20. Notwithstanding my earlier comment about ensuring interpretation is moved out of the Methods into the Discussion, I found the current Discussion was clear, contextualised the results well, and effectively linked results to potential clinical impacts. I had just one query: the authors indicated some TMA cores were lost because of QC check failures. Is this a limitation? Is there any risk the loss of these TMA cores could introduce bias?

Reviewer #2: Very interesting work and could inform subsequent spatial TILs analyses as a prognostic biomarker in breast cancer, although I would argue that nearest neighbor distance is not very novel and in fact it quite a simplistic way to address the spatial biology. Simple could be good (as evidenced by the reported prognostic association), but additional analyses could be included to increase depth/scope and impact, and to convince the reader that NND is in fact optimizing the observed prognostic association.

1) Early-stage breast cancers will have different morphologic features and cellular dispersion patterns, for example, ILC can grow in linear cell bundles, low grade have well defined epithelial nests that often exclude TILs, high grade tumors can be more dispersed/confluent with loss of ductal architecture. I worry that the spatial pattern of the tumor cells will alter the tumor-CD8 NND, and that the spatial pattern of the tumor cells will be correlated with other prognostic features such as tumor grade and subtype, and that this raises the risk of confounding. It would be important to address this in the data somehow. Could you evaluate the spatial patterns of CK+ clustering and see how it correlates with your metric, and see if it influences outcomes of your prediction models. 

2) In relation to comment 1- please evaluate clinical prognostic factors from the cohort that could be added to the multivariate models, to see if the spatial metrics maintain significance. Grade, tumor size/nodal status, ILC v. IDC, age/menopause status, etc. Look for interaction effects to see if there are differences in prognostic impact according to the covariates. 

3) On a similar note, in early-stage breast cancer we have usually have distinct stromal and intraepithelial tissue compartments, and the TILs behave differently across the two compartments, i.e. TILs usually found in stroma but less so in the intraepithelial nests. would your NND metric be influenced unduly in this case by variations in stromal/intraepithelial sampling across the TMAs? You can explore this by quantifying the relative degree of stroma v. epithelia in each sample, and see if this changes the readout.

4) NND and immune cell density can also be influenced by the location of the spatial sample in relation to tumor invasive margin/front. Was this accounted for when the TMA regions were selected? Please describe how in the manuscript. 

5) If you have substantial inter-tumoral heterogeneity you may not accurately capture the mean for the parent specimen from the TMA. It would be interesting to evaluate heterogeneity of your metrics across cores from the same patient. Can you compare the accuracy of your metrics from TMA versus WSI to see if you are getting enough sample ? Does increasing # of TMAs improve the accuracy of your metrics in comparison to the WSI?

6) As you mention in the discussion and elsewhere, CD8/CK NND and CD8 density may interact, i.e. different association of NND with outcome across different CD8 density levels. Have you explored CPH models with interaction terms for the two predictor variables?

7) Density plots in fig 2 are a little confusing to interpret because the cumulative density is split across the subgroups. Could you instead overlay the densities so that for each subgroup the cumulative density is =1, that way you can see how each subgroup differs in terms of their spatial patterns?

8) What about associations of T reg proximity with outcome? T-regs may interfere with CD8s- could you look at associations of CD8-Treg NND with your other metrics, and with outcome? 

9) What is the path forward here for taking this to this clinic? mIF very expensive and not clinically validated. You mention H&E ML TILs classifiers in the discussion. can you analyze your H&E WSI images metrics using these classifiers? Can you look at the same metrics just using H&E? 

Reviewer #3: In this study, Andrew E. Walker et al. have evaluated the prognostic significance of spatial relationships between tumor cells and CD8⁺ T-cells in breast cancer, with a particular focus on estrogen receptor-positive (ER⁺) tumors, which comprise the majority of breast cancer cases but are often overlooked in immune profiling due to low tumor infiltrating lymphocyte (TIL) counts. Utilizing multiplex immunofluorescence and distance-based analyses, the authors introduce two spatial metrics (proximity and consistency) that quantify the immune-tumor interface more precisely than traditional lymphocyte counting. Their findings demonstrate that these spatial metrics are significantly associated with relapse-free survival in both ER⁺ and ER-negative subtypes, and outperform conventional TIL counts in prognostic power. 

The manuscript addresses an important and timely topic, given the growing recognition of spatial immune contexture as a critical component of tumor biology. Current immune quantification methods often fail to capture the full prognostic potential of spatially resolved analyses, and this work represents a step forward in that direction. Overall, the analyses appear well-conceived and promising. However, I offer the following specific comments and suggestions to enhance the rigor and clarity of the manuscript:

1. Please review the REMARK checklist for tumor biomarker studies and include any missing components to improve transparency. For instance, it is unclear whether spatial measurements were conducted blinded to outcome data.

2. The manuscript occasionally shifts between present and past tense in sections such as Image Processing. Consider reviewing and standardizing the tense throughout, ideally using the past tense to describe methods and results.

3. The authors define regulatory T cells as CD8⁺FOXP3⁺, but regulatory T cells are classically CD4⁺FOXP3⁺ and CD8⁻. Figure 1 appears to show many cells labeled as Tregs that express CD8, which would be highly unusual. Could this reflect fluorescence signal bleed-through or misclassification? Have the authors evaluated marker expression within distinct cellular compartments (e.g., nuclear vs. membrane localization)?

4. How was the multiplex immunofluorescence assay validated? The authors are encouraged to review best practices, such as those outlined in the SITC statement (PMID: 32414858), and confirm that staining patterns in the multiplex assay reflect those observed in single-plex IHC.

5. Were lymphocyte counts normalized to the tissue area analyzed? Omitting this step may reduce the accuracy and prognostic relevance of lymphocyte count.

6. Please provide version numbers for all R packages used in the statistical analyses to ensure reproducibility.

7. Because proximity to the nearest lymphocyte is inherently influenced by overall lymphocyte density, a formal correlation analysis between proximity, consistency, and total lymphocyte count would strengthen the results and should be reported and discussed.

8. Consider changing the X-axis of Kaplan-Meier curves from days to years for interpretability and including number-at-risk tables below each curve.

9. Were multivariable Cox regression models performed to adjust for conventional prognostic factors (e.g., tumor size, nodal status, grade)? If not, this should be considered to demonstrate the independent prognostic value of the spatial metrics.

10. Given that regulatory T cells were included in the staining panel, why were no spatial or density-based analyses of Tregs reported? Including such analyses could provide additional insights.

Reviewer #4: Prognostic significance of CD8 T-cell spatial biomarkers in ER+ and ER- breast cancer.

Andrew E. Walker, et al.

This manuscript has a number of significant strengths. Among these is the size of the sample cohort, which lends statistical power to the conclusions. The central question focuses primarily on the relationship between the immune microenvironment and therapeutic outcome in breast cancer, particularly ER+ breast cancer. This is a very significant issue in breast cancer research. Most studies to date have focused on the tumor microenvironment in triple negative breast cancer (HER2+ to a lesser extent), and there is no clear consensus among investigators concerning how the limited number of stromal tumor infiltrating lymphocytes (TILs) contributes to risk of recurrence in ER+/HER2- tumors. 

The authors herein demonstrate a significant spatial component of the immune microenvironment of ER+ tumors. Specifically, they provide convincing evidence that risk of recurrence is associated with the abundance of tumor-proximal CD8+/FOXP3- T-cells. To this end the authors have developed two metrics: proximity and consistency. Proximity is obvious. This parameter incorporates the distance between CD8 cells and tumor cells. Consistency has to do with variance in the distance parameters. Both proximity and consistency were show to be associated with risk of recurrence, particularly in ER+ tumors with relatively low TILs. 

The initial studies were done using tissue microarrays, and a significant strength is the observation that tissue microarray data could be recapitulated in tissue whole mounts from surgical biopsies. This key observation opens the door to clinical assessment of proximity and consistency in the clinical setting. 

The experiments were well designed with appropriate controls and statistical methodology. It could be argued that these observations are limited to a single, broadly defined class of immune cells, CD8+ T-cells. Consideration of other immune cell types is an obvious extension of the present work. Nevertheless, it is the case that this manuscript describes, for the first time to my knowledge, a definitive link between the immune microenvironment and risk of recurrence in ER+ breast cancer. This is clinically a hugely important question, and there is a lot of confusion in the literature. 

In summary, although the studies were somewhat limited in depth, the results establish a very significant association between the immune microenvironment and clinical phenotype in ER+ breast cancer. This observation will quite likely stimulate a more comprehensive analysis of the immune microenvironment in ER+ tumors. 

My only suggestion would be some discussion of what the consistency parameter might mean in terms of CD8+ T-cell distribution. 

Specific review criteria are included below:

Highlights

1. Innovative Spatial Metrics:

o The study introduces two novel spatial biomarkers: Proximity (mean distance from tumor to CD8+ T-cells) and Consistency (variance in these distances). These complement traditional TIL counts and enhance prognostic precision.

2. Focus on ER+ Breast Cancer:

o This is one of the few studies that explores immune infiltration metrics in estrogen receptor-positive (ER+) cancers—an understudied group in the context of immunotherapy response.

3. Large Cohort with Multiplex Imaging:

o Utilizes multiplex immunofluorescence on TMAs from 1,467 patients in the Carolina Breast Cancer Study. This adds robustness and diversity to the dataset.

4. Clinical Relevance:

o Demonstrates that high proximity and consistency correlate with better recurrence-free survival (RFS), particularly in ER+ tumors. These findings may influence treatment selection for a broader patient group.

5. Strong Methodology:

o Rigorous image analysis using QuPath, carefully validated classification thresholds, and robust statistical modeling (Cox regression, Kaplan-Meier, Kruskal-Wallis, Spearman correlation, etc.).

6. Correlations with Established Biomarkers:

o Spatial features (especially proximity) correlate well with established WSI-based TIL metrics and RNA-based immune signatures, supporting their biological validity.

7. Interpretability:

o Clear explanation of metric calculation and biological rationale (i.e., CD8+ T-cells must be in close proximity to tumor cells to induce cytotoxicity).

8. Data Integration:

o Integrates spatial data with molecular classifications (Adaptive-Enriched vs Innate/Quiet, AGI/NGI), providing a multi-modal view of the tumor microenvironment.

Weaknesses & Limitations

1. Limited Scope of Immune Cell Types:

o Only CD8+ and FoxP3+ cells are considered. The exclusion of CD4+ helper T cells, B cells, macrophages, and other immune subsets may oversimplify the TIME.

2. Interpretation of Consistency:

o The metric for Consistency (variance of log-transformed distances) is less intuitive and harder to interpret biologically. Its weaker and less consistent statistical significance also raises concerns about robustness.

3. TMA vs. Whole Slide Imaging (WSI):

o TMAs cover only a small tumor area (1 mm cores), potentially missing heterogeneity present in full tumor sections. While the authors compare TMA and WSI metrics, validation in WSI settings would strengthen generalizability.

4. Potential Overfitting in Cut-point Optimization:

o Supplementary Table 3 shows clear signs of overfitting in training/testing splits when optimizing cut-points for biomarkers, reducing confidence in dichotomization strategies.

5. Data Access Restrictions:

o While CBCS data is available upon request, the requirement for a data use agreement limits replicability and open access principles.

6. Lack of External Validation:

o Findings are derived from a single cohort. Independent validation in external datasets or clinical trials would enhance credibility.

7. Assumption of CD8+ Cell Functionality:

o Assumes that all CD8+ cells are functional cytotoxic T cells, which may not always be the case (e.g., exhausted or anergic phenotypes are not considered).

8. Limited Discussion on Clinical Translation:

o While the implications for immunotherapy are mentioned, the manuscript could further elaborate on how these spatial biomarkers might be integrated into clinical workflows or guide treatment decisions.

Suggestions for Improvement

1. Add multivariate models controlling for clinical covariates (e.g., age, tumor grade, nodal status) to strengthen claims of independent prognostic value.

2. Clarify consistency metric's biological rationale—possibly explore alternative metrics (e.g., Gini index, spatial autocorrelation).

3. Consider evaluating CD8+ functional status (e.g., granzyme B, PD-1 expression) in future studies.

4. Include visual examples or distributions of spatial metrics in different subtypes to better illustrate spatial heterogeneity.

5. Expand on translational relevance—e.g., could spatial metrics predict immunotherapy response in ER+ subtypes?

---

* Please upload any figures associated with your paper as individual TIF or EPS files with 300dpi resolution at resubmission; please read our figure guidelines for more information on our requirements: http://journals.plos.org/plosmedicine/s/figures. While revising your submission, please upload your figure files to the PACE digital diagnostic tool, https://pacev2.apexcovantage.com/. PACE helps ensure that figures meet PLOS requirements. To use PACE, you must first register as a user. Then, login and navigate to the UPLOAD tab, where you will find detailed instructions on how to use the tool. If you encounter any issues or have any questions when using PACE, please email us at PLOSMedicine@plos.org.

* FINANCIAL DISCLOSURES: The funding statement should include: specific grant numbers, initials of authors who received each award, URLs to sponsors’ websites. Also, please state whether any sponsors or funders (other than the named authors) played any role in study design, data collection and analysis, the decision to publish, or preparation of the manuscript. If they had no role in the research, include this sentence: “The funders had no role in study design, data collection and analysis, decision to publish, or preparation of the manuscript.”

* ETHICS: Please provide the approval number.

* In the manuscript text, please indicate: (1) the specific hypotheses you intended to test, (2) the analytical methods by which you planned to test them, (3) the analyses you actually performed, and (4) when reported analyses differ from those that were planned, transparent explanations for differences that affect the reliability of the study's results. If a reported analysis was performed based on an interesting but unanticipated pattern in the data, please be clear that the analysis was data driven. 

* Please state in the Methods section whether the study had a prospective protocol or analysis plan. If a prospective analysis plan (from your funding proposal, IRB or other ethics committee submission, study protocol, or other planning document written before analyzing the data) was used in designing the study, please include the relevant document(s) with your revised manuscript as a Supporting Information file to be published alongside your study and cite it in the Methods section. A legend for this file should be included at the end of your manuscript. If no such document exists, please make sure that the Methods section transparently describes when analyses were planned, and when/why any data-driven changes to analyses took place. Changes in the analysis, including those made in response to peer review comments, should be identified as such in the Methods section of the paper, with rationale.

FIGURES AND TABLES

SUPPLEMENTARY MATERIAL

REFERENCES

---

## [Decision Letter · Decision Letter 2]

29 Aug 2025

Dear Dr. Walker,

Thank you very much for re-submitting your manuscript "Prognostic Significance of CD8 T-cell Spatial Biomarkers in ER+ and ER- Breast Cancer" (PMEDICINE-D-25-01829R2) for review by PLOS Medicine.

Thank you for your detailed response to the reviewers' and editors’ comments. I have discussed the paper with my colleagues, and it has also been seen again by all four original reviewers. The changes made to the paper were mostly satisfactory to the reviewers. As such, we intend to accept the paper for publication, pending your attention to the reviewers' and editors' comments below in a further revision. When submitting your revised paper, please once again include a detailed point-by-point response to the reviewers' and editorial comments.

In revising the manuscript for further consideration here, please ensure you address the specific points made by each reviewer and the editors. In your rebuttal letter you should indicate your response to the reviewers' and editors' comments and the changes you have made in the manuscript. Please submit a clean version of the paper as the main article file. A version with changes marked must also be uploaded as a marked up manuscript file. Please also check the guidelines for revised papers at http://journals.plos.org/plosmedicine/s/revising-your-manuscript for any that apply to your paper. 

We ask that you submit your revision within 1 week (Sep 05 2025). However, if this deadline is not feasible, please contact me by email, and we can discuss a suitable alternative.

Please do not hesitate to contact me directly with any questions (atosun@plos.org).

We look forward to receiving the revised manuscript.

Sincerely,

Alexandra Tosun, PhD

Senior Editor 

PLOS Medicine

plosmedicine.org

Comments from Reviewers:

Reviewer #1: I thank the authors for their thoughtful and comprehensive responses to my comments and note the manuscript is reading very well. In all but two cases, I am happy to accept the authors proposed updates and consider the items finalised. However, I do suggest the authors revisit the two items below. 

10. My understanding is the authors agree with my suggestion not to combine hazard ratios with log-rank test p-values, however this is still present in the abstract "(HR 1.35, 95% CI [0.92,1.98], log-rank p-value = 0.13)". Again, I suggest only presenting Cox model p-values with hazard ratios. 

12. I am afraid I must disagree with the authors on this point. Linear regression absolutely can be used for investigating associative relationships and is not just for "prediction problems" per the authors' response to my review comment. I take the authors' point that the assignment of "outcome" and "covariate" to the two variables of interest can be arbitrary in some cases, but this does not obviate my point that rho only estimates association whereas linear regression provides a meaningful estimate of effect. 

Reviewer #2: I am pleased with the revisions and responses to the plentiful comments. 

Reviewer #3: I think the authors have adequately addressed the comments.

Reviewer #4: - 

Requests from Editors:

GENERAL

* Please confirm that your title complies with to PLOS Medicine's style. Your title must be nondeclarative and not a question. It should begin with main concept if possible. "Effect of" should be used only if causality can be inferred, i.e., for an RCT. Please place the study design ("A randomized controlled trial," "A retrospective study," "A modelling study," etc.) in the subtitle (ie, after a colon).

* Statistical reporting: Please revise throughout the manuscript, including tables and figures.

- Please report statistical information as follows to improve clarity for the reader ""22% (95% CI [13,28]; p</=)"".

- Please separate upper and lower bounds with commas instead of hyphens as the latter can be confused with reporting of negative values.

- Please repeat statistical definitions (HR, CI etc.) for each set of parentheses.

* Please ensure that all abbreviations are defined at first use throughout the text (including statistical abbreviations).

* Please ensure that tables and figures, including those in supplementary files, are appropriately referenced in the main text.

* Please review your text for claims of novelty or primacy (e.g. 'for the first time' or ‘novel’) and remove this language. 

* Please confirm that any use of statistical terms (such as trend or significant) are supported by the data, and if not please remove them. The term trend should be used only when the test for trend has been conducted.

* Please define all acronyms used in each figure or table in its corresponding legend.

* Please revise for use of patient-centered language. Please note that patient-centered language is constructed with the use of post-modified nouns (e.g. 'patients with breast cancer’ (or similar) instead of ‘breast cancer patients’) putting the person first in the sentence structure.

* We feel that there are too many abbreviations used throughout the manuscript, which can be a bit disruptive to the reader. Please revise the main text carefully and significantly reduce the number of abbreviations.

* When revising the manuscript, please keep in mind that, as a general medical journal, we aim to serve a broad audience. Therefore, your manuscript should be accessible to readers who might not be familiar with the contents or background of your study.

* We note that Reviewer #3 recommended using the REMARK checklist. Please provide the completed REMARK checklist. When completing the checklist, please use section and paragraph numbers, rather than page numbers. Please add the following statement, or similar, to the Methods: ""This study is reported as per the Reporting recommendations for tumour MARKer prognostic studies (REMARK) guideline (S1 Checklist)."

* Please confirm that the data availability, financial disclosure, and competing interest statements are in accordance with PLOS requirements, and that you provided the correct, updated versions in the online submission form. You may then remove the statements from the main text.

ABSTRACT

* Please confirm that your abstract complies with our requirements, including providing all the information relevant to this study type https://journals.plos.org/plosmedicine/s/submission-guidelines#loc-abstract

* Please confirm that all numbers presented in the abstract are present and identical to numbers presented in the main manuscript text.

* In the abstract, please include the important dependent variables that are adjusted for in the analyses.

* “Tumor Infiltrating Lymphocytes (TILs) are prognostic in Triple-Negative Breast Cancer (TNBC), but not Estrogen Receptor (ER) positive cancers which comprise 70-80% of breast cancers.” – considering a broad medical audience, we suggest briefly explaining why TILS are not prognostic in ER+ cancers.

* Abstract Background: The final sentence should clearly state the study question.

* “Worse relapse-free survival (RFS) was observed for both ER+ and ER- breast cancers with low proximity and consistency of CD8+ cells.” – we suggest rephrasing to focus on better RFS for high proximity and consistency of CD8+ T-cells.

* “Among ER- breast cancers, proximity had the highest RFS hazard ratio (HR 1.84, 95% CI [1.18,2.87]).” – compared to?

* “spatial characteristics described by proximity and consistency are consistently associated with recurrence irrespective of ER status.” – we believe that this statement might be useful to add to the Abstract findings and/or conclusions. 

AUTHOR SUMMARY

* In the author summary, in the final bullet point of 'What Do These Findings Mean?', please include the main limitations of the study in non-technical language.

METHODS AND RESULTS 

* When reporting ages, please include a unit, such as ‘years’. Please revise throughout.

* Please include the ethics approval number.

* Figure 1: How was race/ethnicity defined and by whom? Why was race/ethnicity considered important in this study and what it is believed to represent [eg, are SES or genetic differences being attributed to race/ethnicity?]

* Please clarify at what temperatures the different IF staining steps were conducted.

* Please note that you switch between different descriptions of CD8+ T-cells (including CD8+ cells, CD8+ T-cells, CD8+ lymphocyte). Please revise and use a consistent format.

* Please note that you switch between ‘recurrence-free survival’ and ‘relapse-free survival’. Please revise and use a consistent format.

* Please ensure that where relevant figures include 95% CIs.

* Please show graph axes beginning at zero. If this is not possible, please show a break in the axis.

* In the Kaplan-Meier curve(s) please provide the number at risk for each time interval.

* Please specify the variables controlled for in all relevant Tables.

General Editorial Requests

---

## [Editor Report · Decision Letter 3]

19 Sep 2025

Dear Dr Walker, 

On behalf of my colleagues and the Academic Editor, Ricky W Johnstone, I am pleased to inform you that we have agreed to publish your manuscript "Prognostic Significance of CD8+ T cell Spatial Biomarkers in ER+ and ER- Breast Cancer: A Retrospective Cohort Study" (PMEDICINE-D-25-01829R3) in PLOS Medicine.

I appreciate your thorough responses to the reviewers' and editors' comments throughout the editorial process. We look forward to publishing your manuscript, and editorially there is only one remaining point that should be addressed prior to publication. We will carefully check whether the request has been addressed. If you have any questions or concerns regarding the final request, please feel free to contact me at atosun@plos.org.

Please see below the minor point that we request you respond to:

* Please note that you have not uploaded the REMARK checklist as Supporting Information file.

Before your manuscript can be formally accepted you will need to complete some formatting changes, which you will receive in a follow up email (including the editorial point above). Please be aware that it may take several days for you to receive this email; during this time no action is required by you. Once you have received these formatting requests, please note that your manuscript will not be scheduled for publication until you have made the required changes.

PRESS

Sincerely, 

Alexandra Tosun, PhD 

Senior Editor 

PLOS Medicine